# Assessment of Glucose Lowering Medications’ Effectiveness for Cardiovascular Clinical Risk Management of Real-World Patients with Type 2 Diabetes: Targeted Maximum Likelihood Estimation under Model Misspecification and Missing Outcomes

**DOI:** 10.3390/ijerph192214825

**Published:** 2022-11-11

**Authors:** Veronica Sciannameo, Gian Paolo Fadini, Daniele Bottigliengo, Angelo Avogaro, Ileana Baldi, Dario Gregori, Paola Berchialla

**Affiliations:** 1Centre for Biostatistics, Epidemiology and Public Health, Department of Clinical and Biological Sciences, University of Turin, Regione Gonzole 10, 10043 Orbassano, Italy; 2Department of Medicine, University of Padova, 35128 Padova, Italy; 3Unit of Biostatistics, Epidemiology and Public Health, Department of Cardiac, Thoracic, Vascular Sciences and Public Health, University of Padova, 35128 Padova, Italy

**Keywords:** cardiovascular risk management, type 2 diabetes, real-world data, model misspecification, missing outcome data, TMLE, propensity score

## Abstract

The results from many cardiovascular (CV) outcome trials suggest that glucose lowering medications (GLMs) are effective for the CV clinical risk management of type 2 diabetes (T2D) patients. The aim of this study is to compare the effectiveness of two GLMs (SGLT2i and GLP-1RA) for the CV clinical risk management of T2D patients in a real-world setting, by simultaneously reducing glycated hemoglobin, body weight, and systolic blood pressure. Data from the real-world Italian multicenter retrospective study Dapagliflozin Real World evideNce in Type 2 Diabetes (DARWINT 2D) are analyzed. Different statistical approaches are compared to deal with the real-world-associated issues, which can arise from model misspecification, nonrandomized treatment assignment, and a high percentage of missingness in the outcome, and can potentially bias the marginal treatment effect (MTE) estimate and thus have an influence on the clinical risk management of patients. We compare the logistic regression (LR), propensity score (PS)-based methods, and the targeted maximum likelihood estimator (TMLE), which allows for the use of machine learning (ML) models. Furthermore, a simulation study is performed, resembling the structure of the conditional dependencies among the main variables in DARWIN-T2D. LR and PS methods do not underline any difference in the effectiveness regarding the attainment of combined CV risk factor goals between the two treatments. TMLE suggests instead that dapagliflozin is significantly more effective than GLP-1RA for the CV risk management of T2D patients. The results from the simulation study suggest that TMLE has the lowest bias and SE for the estimate of the MTE.

## 1. Introduction

Coronary heart disease, cerebrovascular disease, peripheral arterial disease of an atherosclerotic origin, and heart failure are the leading causes of morbidity and mortality in diabetic patients, resulting in an estimated $37.3 billion in cardiovascular (CV)-related spending per year. Common conditions coexist in type 2 diabetes (T2D) patients, such as hypertension or dyslipidemia, which make the CV risk management of T2D individuals more challenging [1,2].

Many CV outcome trials in the last decades have suggested that glucose lowering medications (GLMs) can be effective for the CV risk management of T2D patients. In particular, it was found that certain specific sodium-glucose cotransporter-2 inhibitors (SGLT2i) [3,4] and glucagon-like peptide-1receptor agonists (GLP-1RA) [5,6,7], which are two classes of GLMs, are both safe and effective for CV risk management, by addressing risk factors associated with adverse cardio–renal outcomes in T2D patients.

In general, SGLT2is are supposed to be less effective at lowering glucose than GLP-1RA; meanwhile, a network meta-analysis suggests that there are no differences in the glycemic effects of GLP-1RA or SGLT2i when added after dual-therapy failure [8]. Furthermore, some trials showed that both GLP-1RA and SGLT2i are useful for managing other CV-associated risk factors, which typically occur in T2D patients, such as the increase in body weight (BW) and systolic blood pressure (SBP) [9,10]. However, the STENO-2 study focuses on the importance of dealing with composite outcomes, i.e., the simultaneous management of multiple CV risk factors [11].

However, today, the number of RCTs that compare SGLT2i and GLP-1RA considering composite outcomes that deal simultaneously with an ensemble of CV risk factors, such as glycated hemoglobin (HbA1c), BW, and SBP, are lacking. In this case, real-world data are very useful for integrating knowledge with medium-level evidence, which can be helpful to contribute to the clinical decision-making process [12,13].

To address this gap, in our previous work [12], we conducted a retrospective real-world study to compare the effectiveness of dapagliflozin (SGLT2i) and GLP-1RA in T2D patients, evaluating the proportion of patients with a simultaneous reduction in CV-related risk factors (HbA1c > 0.5%, BW > 2 kg, and SBP > 2 mm Hg), and no differences were found. However, we had to deal with the typical issues related to real-world observational studies, such as missingness (we deleted observations with missing outcome data), model misspecification, and the absence of randomization in the treatment assignment.

In fact, in observational biomedical research, the evaluation of the marginal (i.e., at the population level) treatment effect (MTE) is made more challenging by nonrandomized treatment assignment and missingness in both covariates (X) and in outcome (Y) measures. Treatment groups often differ in baseline characteristics, and an appropriate statistical method that adjusts for confounding variables and considers the missingness mechanism both in the covariates and outcome is necessary to minimize the bias in MTE estimates so as to improve the causal inference.

This study aims to compare in a real-world setting the effectiveness of dapagliflozin (SGLT2i) and GLP-1RA in the CV risk management of T2D patients by simultaneously controlling CV associated risk factors (HbA1c, BW, and SBP).

The results from different statistical approaches are compared to limit the size of the bias in the estimate of the MTE due to model misspecification and non-randomized treatment assignment. In special cases, the missing dichotomous outcome data are generated under a missing not at random (MNAR) or missing at random (MAR) mechanism [14]. Both real-world and simulated data tailored to the case study are analyzed.

## 2. Materials and Methods

### 2.1. Case Study with Real-World Data

We analyzed real-world data from the Italian multicenter retrospective nationwide Dapagliflozin Real World evideNce in Type 2 Diabetes (DARWIN-T2D) study, conducted at 46 diabetes specialist outpatient clinics in Italy. One of the aims of DARWIN-T2D is to describe the baseline clinical features in diabetic patients initiated on dapagliflozin compared with those who initiated on a comparator GLM, such as GLP-1RA, and to compare the changes in glycemic efficacy parameters [15]. The secondary objectives were to compare the effectiveness of dapagliflozin and other GLMs, including GLP-1RA, in the CV risk management of diabetic individuals. T2D patients initiated on dapagliflozin or GLP-1RA were compared to evaluate the proportion of patients with a simultaneous reduction in HbA1c > 0.5%, BW > 2 kg, and SBP > 2 mm Hg [12]. More details about the DARWIN T2D study can be found in previous publications [12,15].

The MICE algorithm [16] was applied to impute the missing covariate data, and five imputed datasets were obtained. Only covariates with less than 40% missing data were included as predictors in the imputation process, including the observed outcome values [17,18,19,20]. The outcome variable was not imputed, because it has been shown that when missingness is in the outcome variable, complete case (CC) and multiple imputation (MI) strategies [21] lead to similar results, with a low bias only when the missingness mechanism is missing completely at random (MCAR) or missing at random (MAR); however, in our case study, we cannot a priori exclude a missing not at random (MNAR) mechanism on the outcome [22,23]. In fact, subjects with observed and missing outcome data significantly differed in terms of age, waist circumference, fasting glucose, total and LDL cholesterol, eGFR, insulin associated therapy, and ACEi/ARBs therapy [12].

In each imputed dataset, we estimated the propensity score (PS) model via the logistic regression (LR) approach, considering the following baseline covariates of age, sex, duration of diabetes, BW, body mass index (BMI), fasting plasma glucose (FPG), HbA1c, systolic, and diastolic pressure (SBP and DBP, respectively), total and HDL cholesterol, triglycerides, eGFR, insulin and metformin therapy, micro-angiopathy, and macro-angiopathy.

The multiple drugs taken by patients were not considered in this analysis, because in our previous work [12], we performed a sensitivity analysis where the number of prior GLM drug classes was added as a covariate in the PS models, but we observed that there were no differences in the results. Furthermore, the median number and range of prior GLM classes were superimposable in the two treatment groups.

PS matching was performed with the nearest 1:1 ratio without replacement and with a caliper of 0.15 standard deviations of the distribution of the PSs on the logit scale [24].

Outcome analyses were performed on each imputed database, and the results were pooled together following Rubin’s rules [25] and the within approach [26].

### 2.2. Simulation Study

Data were simulated by defining a directed acyclic graph (DAG), as shown in Figure 1, using the simcausal R package (R version 3.2.0) [27,28].

The simulation process was set up to resemble the relationships between the main variables in the case study (DARWIN-T2D), i.e., sex (W_1_), age at diagnosis (W_2_), BMI (W_3_), LDL cholesterol (W_4_), insulin use (W_5_), and macro angiopathy (W_6_). We constructed a Bayesian Network (BN) on variables extracted from the DARWIN-T2D dataset obtaining the conditional probability distributions, which reflect the dependencies among these variables. Then, the Peter–Clark stable algorithm with 100-fold bootstrap was applied for the structural learning of the BN [29]. Finally, a more robust BN was obtained by averaging the 100 BNs learned and considering only the relationships between the variables present at least 95% of the time [30]. Thus, using such dependencies relations extracted from DARWIN-T2D and the summary statistics of the main variables of the case study, we considered the following simulation process:W_1_ ~ Bernoulli (0.6);W_2_ ~ Gaussian (mean = 60, sd = 8);W_3_ ~ Gaussian (mean = 35, sd = 6);W_4_ ~ Gaussian (mean = 90 if W_1_ = 1 and mean = 97 if W_1_ = 0, sd = 30),W_5_ ~ Bernoulli (plogis (−2 + 0.05 × W_2_));W_6_ ~ Bernoulli (plogis (−12 + 0.50 × W_3_ − W_1_));Z ~ Bernoulli (plogis (−2 + 0.05 × W_3_ − 0.20 × W_5_ + 0.10 × W_6_));Y ~ Bernoulli (plogis (−3 + Z − 0.05 × W_2_ + 0.05 × W_4_ − 0.80 × W_1_ − 0.20 × W_1_ × W_2_)), where plogis is the inverse logit function: 1/log[p/(1 − p)].

In the case study, because subjects with observed and missing outcome data significantly differed in many covariates [12] and the use of the less studied and deepened mechanism and a lack of knowledge on how to deal with it, we simulated an MNAR mechanism on the outcome using two different frequencies: 20% and 40%. In more detail, the probability of missingness in Y depends on the Y value itself: if Y = 1, the probability of missingness is 70%. Because it was not possible to determine what kind of missing outcome mechanism was underlying the case study data, the MAR mechanism was also simulated as a sensitivity analysis, which was performed using a multivariate amputation procedure. For each unit, a weighted sum score was computed via a linear regression equation for each covariate with the same weight, excluding the outcome variable. Then, each patient was assigned a probability of having missing outcome data based on his/her weighted sum score, following a right-tailed logistic distribution function, i.e., subjects with a higher weighted sum score had a higher probability of having missing outcome data [31]. The results of the MAR scenarios are included in the Appendix A.

Missingness mechanisms were simulated through the ampute function in the mice R package [31].

In the simulation study, we performed the comparisons in the most common real-world situation: both the treatment and outcome models were misspecified. More specifically, in the Z model, W_2_ and W_6_ were included as covariates, and W_3_ and W_4_ were included in the Y model, as follows:$$Y\sim \alpha_{0}Z+\alpha_{1}W_{3}+\alpha_{2}W_{4}$$
$$Z\sim \alpha_{3}W_{2}+\alpha_{4}W_{6}$$

The outcome and treatment models (Y and Z, respectively) that we estimated were in fact misspecified, because we included in the estimation of the outcome model (Y) covariate W_3_, which was not present in the true outcome model (Y), and we did not include W_1_, W_2_, and the interaction term W_1_ × W_2_, which were instead present in the true model.

In the same way, in the treatment model (Z), we included in the estimation of the model the covariate W_2_, which was not in the true model, and we omitted W_3_ and W_5_, which were instead in the true model.

We estimated the PS models via the LR approach. PS matching was performed with the nearest 1:1 ratio without replacement and with a caliper of 0.15 standard deviations of the distribution of PS on the logit scale [24].

The true marginal OR(mOR) value (1.66) was evaluated on 5,000,000 observations generated through DAG in Figure 1.

Sensitivity to the sample size was addressed by performing the analyses with both *n* = 1000 and *n* = 5000.

Overall, 1000 simulations were performed to estimate the mean bias measure, the standard error (SE) size, and the 95% nominal coverage (NC) intervals. The bias was defined as the differences between the true mOR and the mean value of the estimates of OR obtained in all of the simulations, i.e.,
$$\mathrm{bias}=\mathrm{true}\left(\mathrm{OR}\right)-E\left(\hat{\mathrm{OR}}\right)$$

Statistical analyses were performed using R version 3.4.5 [32].

### 2.3. Statistical Methods

In this study, we compared, in both real-world and simulated settings, the marginal odds ratio (mOR) in order to estimate the effectiveness of GLMs in the CV risk management of T2D patients, by controlling CV-related risk factors, according to the following methods: (i) logistic regression for covariate adjustment (LR), (ii) PS adjustment, (iii) PS matching, (iv) inverse probability of treatment weighting (IPTW), and (v) the targeted maximum likelihood estimator (TMLE), which are the most widely used approaches in the literature [33]. In the following sections, more details about the statistical methods are provided.

#### 2.3.1. Logistic Regression for Covariate Adjustment (LR)

When the outcome is dichotomous, the most traditional method to estimate the MTE, while controlling for confounders, is the logistic regression (LR) model. However, many assumptions must be satisfied: the model should be fitted correctly, all the confounders must be measured, the observations should be independent of each other, no multi-collinearity among the independent variables is allowed, and the relationship between the independent variables and the log odds must be linear. Often, in real-world data, these assumptions are not verified or are not testable, resulting in a high probability of model misspecification, with a consequent increase in bias in the treatment effect estimate.

Furthermore, the LR model approximates the estimate of mOR with the estimate of conditional OR (cOR), which has to be interpreted at the subject level, assuming homogeneity in the treatment effect for subjects with the same observed covariates, without taking into account possible heterogeneity in the treatment [34]. cOR is more reliable in a randomized controlled setting, where, generally, problems related to unmeasured confounding are controlled by the randomization itself [35].

#### 2.3.2. Propensity Score Based Methods: Adjustment, Matching, IPTW

mOR, introduced in 1974 by Rubin, is instead interpreted at the population level and takes advantage of the potential outcome framework [36]. Each subject has two potential outcomes: the outcome Y1 he/she would have experienced if he/she had been treated (Z = 1), and the outcome Y0 he/she would have had if he/she had not been treated (Z = 0). For each subject, it was possible to observe only one realization of the outcome.

Rosenbaum and Rubin introduced PS-based methods to estimate the mOR in the potential outcomes framework [37]. PS is defined as the probability to be assigned to a treatment (Z, a binary variable), conditioned on baseline covariates **W**, mathematically P (Z = 1|**W**). PS allows for the balance of baseline covariates between two different treatment groups via matching, inverse probability of treatment weighting (IPTW), or stratification [38,39].

However, PS methods are sensitive to the misspecification of both treatment and outcome models [40]. It has been shown that if the treatment and/or outcome model is misspecified, the mOR estimate is biased in the direction of cOR [35]. In other words, PS methods require the absence of unmeasured confounders or, equivalently, that the randomization assumption is satisfied, which means that given the baseline covariates **W**, the treatment Z is independent of the potential outcomes Y1 and Y0 [37]. Additionally, PS methods require that the positivity assumption is satisfied; that is, for any values of the baseline covariates **W**, the probability of being treated is nether 0 nor 1: 0 < P (Z = 1|**W**) < 1 for all **W** [37]. Often, in real-world data, these assumptions cannot be verified, and a biased MTE estimate could be obtained via PS methods.

#### 2.3.3. Targeted Maximum Likelihood Estimator (TMLE)

TMLE is a maximum likelihood estimator based on the G computation estimator, introduced in 2006 by Van der Laan and Rubin. TMLE is a semiparametric double-robust method that improves the chances of correctly specifying the treatment and outcome model, taking advantage of flexible estimation via nonparametric machine learning (ML) methods [28]. Double-robust methods were developed to minimize the impact of model misspecification that has heavier consequences on biased causal inference [41,42,43,44]. It computes initial estimates of the outcome regression Q^0^(Z, **W**) = E (Y|Z, **W**, Δ = 1) and PS models based on complete observations only. Δ represents the model regarding the missingness mechanism on the outcome Y. Δ = 1 indicates the observations for which the outcome is not missing, meanwhile Δ = 0 represents observations with missing outcomes. Then, the initial outcome regression model is updated, by constructing a logistic model
$${\mathrm{logit}\;Q}_{\;}^{0}\left(\varepsilon \right)\left(Y=1|Z,\;W,\Delta \;=1\;\right)={\;\mathrm{logit}\;Q}_{\;}^{0}\left(Y=1|Z,\;W,\Delta \;=1\;\right)+\;\varepsilon H\left(Z,\;W\right)$$
where H (Z, **W**) are “clever covariates” and ε is computed via the maximum likelihood procedure. The “clever covariates” take advantage of the information contained in the PS model and in the missingness mechanism on the outcome model P (Δ = 1|Z, **W**) as follows:$$H\left(Z,\;W\right)=\frac{1}{P(\Delta \;=1|Z,\;W)}\;\left(\frac{Z}{P\left(Z=1|W\right)}-\frac{1-\;Z}{P\left(Z=0|W\right)}\;\right)$$

The update is performed in the potential outcome framework both for Z = 0 and Z = 1 for all subjects. Finally, the empirical mean of the predicted probabilities of the counterfactual outcome is computed, and mOR is obtained. The efficient influence function is then used to compute the SE and the Wald-type 95% confidence interval. Both the initial and the updated estimates can be estimated via the super learner (SL) algorithm, which is a statistical approach based on ML ensemble methods that finds the optimal combination of a collection of algorithms to minimize the cross validated risk, i.e., the prediction error computed in the cross-validation framework. A mathematically proven theorem states that the SL algorithm performs asymptotically, as well as the oracle selector, i.e., the best candidate between learner algorithms [45,46,47].

Furthermore, TMLE allows for simultaneously dealing with missing outcome data and model misspecification that, in the case of MAR or MNAR in the outcome, can have a stronger contribution for obtaining biased causal inference [41,42]. A procedure of inverse probability weighting (IPW) to deal with missing outcome data is intrinsically implemented in the TMLE algorithm. More details about TMLE can be found in [33,46,48].

Two analyses based on the TMLE were performed. In the first (TMLE1), the super learner (SL) algorithm includes only the default algorithms (main terms LR, stepwise forward and backward model selection, and main terms LR with interaction terms). In the second (TMLE2) one, the SL algorithm includes the following algorithms: main terms LR, stepwise forward and backward model selection, main terms LR and interaction terms, stepwise forward and backward model selection with interaction terms, generalized additive models (GAMs), random forest (RF), and recursive partitioning and regression trees (RPART).

TMLE analyses are performed under two different strategies to handle missing outcome data: the complete case (CC) approach, i.e., excluding observations with missing outcome (TMLE_CC), and the IPW approach, which is included in the TMLE itself (TMLE_IPW).

## 3. Results

### 3.1. Case Study with Real-World Data

From a population of 281,217 patients with T2D collected in DARWIN-T2D, longitudinal data were collected for 2484 patients who initiated Dapagliflozin and 2247 who initiated a GLP-1RA medication. A follow up examination was available for 830 patients in the Dapagliflozin group and 811 patients in the GLP-1RA group. Composite outcome data were available for 473 patients who initiated Dapagliflozin and for 336 patients who initiated GLP-1RA. Therefore, the missingness of the outcome variable was 49%. Furthermore, in a previous work [12] in which we made the analyses only on subjects with an observed outcome, we noticed that patients with observed outcome data differed systematically in several covariates (such as age, waist circumference, fasting glucose, total and LDL cholesterol, eGFR, insulin associated therapy, and ACEi/ARBs therapy) from patients with missing outcome data. Thus, the missingness mechanism on the outcome data did not seem to be MCAR, and CC analyses could lead to significant bias in the estimation of the MTE of the GLMs in controlling the CV-related risk factors. PS matching analysis was performed between 229 subjects in each group. A good balance in the covariates was achieved (all standardized differences were less than 0.10). More details about the PS analyses can be found in [12].

Table 1 shows the results of the effect of Dapagliflozin, compared with GLP1RA, obtained via different statistical approaches. ORs < 1 indicate that attaining the composite outcome occurred less frequently in the Dapagliflozin group than in the GLP-1RA group. LR and PS methods with the CC analysis yielded similar results: they did not underline any differences between the two GLMs in controlling CV-related risk factors. On the other hand, TMLE2 (IPW) showed an OR = 1.35, with a significant 95% confidence interval (CI) (1.04–1.73).

In Table 2, the coefficients of the algorithms selected by SL in TMLE2 (IPW) for the RW case study are reported. Only the LR model with interaction terms was never selected.

### 3.2. Simulation Study

In Table 3, the results of the simulation study are reported, with different amounts of MNAR outcome data and different sample sizes (*n* = 1000 and *n* = 5000).

Regardless of the amount of missingness on outcome data and sample size, TMLE is the approach with the smallest bias and SE. The results show that including the missingness mechanism on the outcome in TMLE (TMLE 2 IPW) improves the OR estimation. The same conclusions were achieved by simulating a MAR mechanism on the outcome, as shown in Appendix A.

There are no relevant differences in terms of the 95% NC intervals in the MNAR scenario; however, they were higher for TMLE approaches in the MAR scenario (Appendix A).

## 4. Discussion

Many CV outcome trials suggest that GLMs are effective for the CV risk management of T2D patients by simultaneously controlling the CV-related risk factors, such as HbA1c, BW, and SBP. However, RCTs that consider such risk factors simultaneously are still few, and real-world studies can help to fill this gap and can be helpful to contribute to the clinical decision-making process. Observational data are often used to study causal questions, even if their estimates are more biased than clinical trials. However, Hernan highlights the importance of having multiple studies with imprecise estimates rather than having no study at all. Then, after several studies become available, it is important to meta-analyze them to provide a more precise pooled effect estimate [49].

Real-world data availability is constantly growing, but advanced statistical approaches are needed to address real-world data-related limitations, such as the absence of randomization, the presence of measured and unmeasured confounders, and the high percentage of missing data, both in the outcome and covariates.

In this work, we compared traditional approaches, such as LR- and PS-based methods, with an advanced ML approach (TMLE), to estimate the effectiveness of GLMs in simultaneously controlling CV-related risk factors (i.e., HbA1c, BW, and SBP). In the real-world case study (DARWIN-T2D), traditional approaches showed no difference between the two treatments. Contrariwise, TMLE pointed out an opposite association compared with the estimates obtained using traditional methods. In fact, according to the TMLE, dapagliflozin simultaneously reduced the CV-related risk factors (HbA1c, BW, and SBP) significantly more than GLP-1RAs. As no RCTs provide a background for this clinically relevant comparison, our results have particularly interesting therapeutic implications for routine clinical practice. Missingness patterns in real-world data may be driven by the characteristics of the different therapies being compared, thereby affecting the outcome comparison. The reversing of the OR could be explained by the fact that, often, LR- and PS-based methods give a biased estimate of the mOR that stretches in the direction of the cOR [50,51] when the model is misspecified. When the conditional and marginal treatment effects do not coincide and they are in opposite directions [52], we refer to this situation as the non-collapsibility of the OR [36]. Furthermore, because of the non-collapsibility of the OR, even a correctly specified LR model generally does not produce estimates of the marginal treatment effect [28].

Furthermore, TMLE can gather interactions between covariates and nonlinearity through the SL algorithm, which could contribute to the change in the direction of the OR. In fact, applying the TMLE without the intervention of the SL algorithm but using the GLM approach only for the Y, Z, and Δ models, we obtain a weaker OR with a non-statistically significant 95% CI (OR = 1.11; 95% CI 0.79–1.69).

Other studies used simulation methods to compare the properties of TMLE to those of other statistical estimators. For example, Pang and colleagues [33] evaluated the performance of TMLE compared with an IPW estimator of the MTE of post-myocardial infarction statin use considering the 1-year risk for all-cause mortality from the Clinical Practice Research Datalink. Their simulation study showed that when TMLE and IPW estimators used the same treatment model specification, they may perform differently due to differential sensitivity to practical positivity violations. However, they observed that TMLE, which is a doubly robust estimator, showed an improved performance with richer specifications than the outcome model.

Furthermore, Schuler et al. [53] performed a simulation study to compare the methods under parametric regression misspecification, and their results highlight TMLE’s property of double robustness. Moreover, their simulation study demonstrated that incorporating ML techniques in SL may outperform parametric regression in observational data settings.

In this study, we also observed that TMLE took advantage of the non-parametric algorithms included in the SL to address model misspecification. Still, unlike previous studies, we looked in detail at the missingness mechanisms in the outcome data. In particular, we focused on MNAR and MAR missingness mechanisms on the outcome, as it is still unclear how to deal with this and there may be relevant consequences of model misspecification, typically occurring in real-world observational studies [41,42,54].

In [55,56], the authors showed that TMLE implemented with the SL algorithm had the lowest bias compared with misspecified-parametric-model-based methods; however, a comparison of how the mechanism of outcome missingness affects their performance is still lacking.

In our simulation study tailored to the real-world dataset, TMLE was confirmed to have the lowest bias and SE, even with a large amount of missing outcome data, under MNAR and MAR mechanisms. Additionally, we found that TMLE performed better than the CC approach if the missingness mechanism was included in the OR estimation process, confirming that CC must be used only when MCAR or MAR mechanisms are present [22]. Different missingness mechanisms on the outcome were addressed, and more attention was given to MNAR, which is still poorly studied and guidelines on how to deal with it are still lacking. Furthermore, the consequences of model misspecification are more relevant with MNAR than with MAR or MCAR [41,42], so it is important not to neglect this aspect and not to a priori exclude the possibility of having MNAR missing outcome data [57].

The 95% NC intervals from the different methods were similar when the MNAR mechanism on Y was present, while the 95% NC was higher for the TMLE when a MAR mechanism was considered.

One limitation of the TMLE was its complexity, intensified by using the SL algorithm for constructing the outcome, treatment, and missingness mechanism on the outcome models. This complexity entailed a greater computational effort compared with more traditional approaches, but was exceeded by the superiority of the TMLE in terms of robustness and stability.

A limitation of this study is that only a few scenarios were considered in the simulation study. This choice was justified by providing a simulation scheme as close as possible to the real-world data characteristics of the case study, taking advantage of Bayesian networks. However, we made some simulations also varying some settings, such as the amount of missingness in the outcome or the sample size, in order to test our results’ stability and generalizability, obtaining promising insights.

The results of this study suggest that, in observational studies, TMLE could simultaneously deal with misspecification and missingness on outcome data, even under MNAR (or MAR) scenarios, outperforming both the LR model and PS methods in terms of bias, SE, and 95% NC intervals. Traditional approaches require lots of assumptions to be satisfied, and they are more suitable in the presence of MCAR or MAR mechanism on the outcome; however, it is usually difficult to identify the missingness mechanism underlying the data, and a priori excluding an MNAR mechanism [55] could amplify the consequences of the model misspecification.

Our recommendation is not a priori excluding MAR or MNAR schemes, and to use simulations to identify which approach is the more adapt to that specific situation.

## 5. Conclusions

In conclusion, as RCTs providing a background for the comparison of the effectiveness of SGLT2i and GLP-1RA in the simultaneous management of CV-related risk factors in T2D patients are still lacking, the results of this study can have particularly interesting therapeutic implications for routine clinical practice and can contribute to the clinical decision-making process for the CV risk management in T2D individuals.

Furthermore, our study confirms that TMLE has appealing statistical properties, i.e., it can simultaneously deal with model misspecification through advanced ML algorithms used by SL and with missing outcome data, even under the MNAR mechanism, which are typical issues in real-world studies that evaluate the effectiveness of medications.

## Figures and Tables

**Figure 1 ijerph-19-14825-f001:**
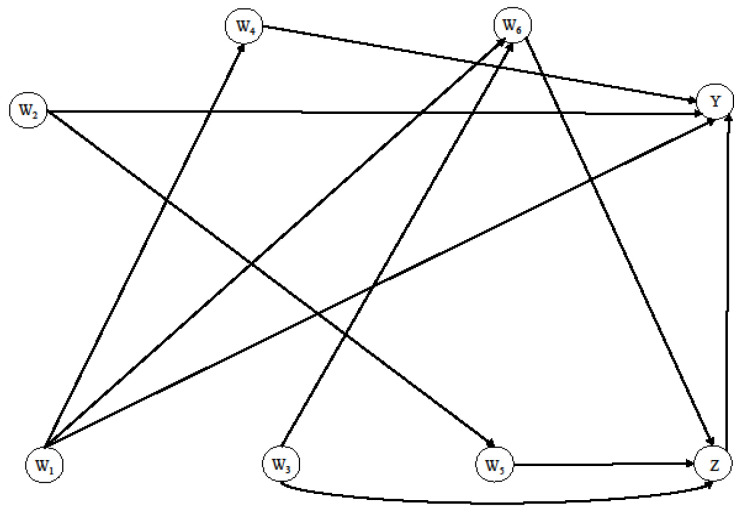
DAG of the simulated data.

**Table 1 ijerph-19-14825-t001:** The results of the DARWIN T2D study.

Method	OR (95% CI)
LR	0.82 (0.53−1.27)
PS matching	0.86 (0.53−1.41)
PS covariate	0.81 (0.52−1.26)
IPTW	0.85 (0.54−1.34)
TMLE 1 (CC)	1.33 (0.91−1.96)
TMLE 2 (CC)	1.53 (1.09−2.14)
TMLE 1 (IPW)	1.34 (0.95−1.91)
TMLE 2 (IPW)	1.35 (1.04−1.73)

Dapagliflozin vs. GLP 1RA. OR = odds ratio; 95% CI = 95% confidence interval; LR = logistic regression; PS = propensity score; IPTW = inverse probability of treatment weighting; TMLE = targeted maximum likelihood estimator; CC = complete case; IPW = Missing Outcome Data. TMLE1: LR with main terms only, LR with stepwise model selection procedures, LR with interaction terms; TMLE2: LR with main terms only, LR with stepwise procedures, LR with interaction terms, LR with interaction terms with stepwise model selection procedures, generalized additive models (GAM), random forest (RF), and recursive partitioning and regression trees (RPART).

**Table 2 ijerph-19-14825-t002:** Coefficients of the algorithms selected by the super learner algorithm in TMLE2 (IPW) for the DARWIN T2D study.

	Model		
	Outcome (Y)	Treatment (Z)	Missingness (Δ)
**Super Learner algorithms**			
LR	0	0	0.09
Step	0.51	0.19	0
Step + interaction terms	0.12	0.10	0
LR + interaction terms	0	0	0
GAM	0.31	0	0.21
RF	0.03	0.53	0
RPART	0.03	0.18	0.70

LR = logistic regression; Step = LR with stepwise forward and backward model selection; GAM = generalized additive model; RF = random forest; RPART = recursive partitioning and regression tree.

**Table 3 ijerph-19-14825-t003:** The results of the simulation study with different scenarios and the MNAR mechanism on the outcome.

	LR	PS Matching	PS Covariate	IPTW	TMLE1 (CC)	TMLE2 (CC)	TMLE1 (IPW)	TMLE2 (IPW)
SCENARIO 1: 20% MNAR on Y, *n* = 1000								
OR	1.89	1.90	1.88	1.84	1.79	1.79	1.75	1.75
Bias	0.23	0.25	0.23	0.18	0.13	0.13	0.09	0.09
SE	0.67	0.84	0.68	0.66	0.58	0.58	0.58	0.57
95% NCI	94.5	93.9	94.0	94.5	94.5	92.4	93.9	91.6
SCENARIO 2: 40% MNAR on Y, *n* = 1000								
OR	2.06	2.07	2.05	2.00	1.97	1.98	1.92	1.94
Bias	0.40	0.41	0.39	0.34	0.31	0.32	0.26	0.28
SE	1.30	1.31	1.30	1.23	1.20	1.18	1.15	1.17
95% NCI	94.2	96.0	94.6	95.5	94.9	92.4	94.2	91.2
SCENARIO 3: 20% MNAR on Y, *n* = 5000								
OR	1.78	1.77	1.77	1.74	1.70	1.70	1.66	1.66
Bias	0.12	0.11	0.11	0.08	0.04	0.04	0.002	0.006
SE	0.25	0.28	0.25	0.25	0.22	0.22	0.22	0.22
95% NCI	92.0	93.5	92.7	94.6	95.6	94.2	94.6	93.4
SCENARIO 4: 40% MNAR on Y, *n* = 5000								
OR	1.78	1.79	1.78	1.74	1.73	1.73	1.68	1.68
Bias	0.12	0.13	0.12	0.08	0.07	0.07	0.02	0.02
SE	0.37	0.43	0.37	0.36	0.35	0.35	0.35	0.35
95% NCI	95.2	95.6	95.1	95.7	95.8	94.8	96.1	95.2

OR = odds ratio; SE = standard error; 95% NC = 95% nominal coverage interval; MNAR = missing not at random; n = sample size; LR = logistic regression; PS = propensity score; IPTW = inverse probability of treatment weighting; TMLE = targeted maximum likelihood estimator; CC = complete case; IPW = inverse probability weighting. TMLE1: LR with main terms only, LR with stepwise model selection procedures, LR with interaction terms; TMLE2: LR with main terms only, LR with stepwise procedures, LR with interaction terms, LR with interaction terms with stepwise model selection procedures, generalized additive Models (GAM), random forest (RF), and recursive partitioning and regression trees (RPART).

## Data Availability

Scripts are available upon request.

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
