# Peer review of "Assessment of Glucose Lowering Medications’ Effectiveness for Cardiovascular Clinical Risk Management of Real-World Patients with Type 2 Diabetes: Targeted Maximum Likelihood Estimation under Model Misspecification and Missing Outcomes"

_ijerph, 2022, doi:10.3390/ijerph192214825_

Round 1

Reviewer 1 Report

Veronica Sciannameo performed a real-world analysis of T2DM diseases and cardivascular risks. This is an interesting multi-center study with some convincing data. Here are the comments from the reviewer:

1. The total number of patients included and the number of centers should be included.

2. The multiple drugs patients taken should be included and analyzed.

3. Even though the authors performed real-time study, ethical approval is still needed.

4. There are only three tables with few data. The author should explore more data. 

Author Response

Dear Reviewer,

Thank you for your comments, which were of great help in the process of revision of the manuscript ijerph-1928122 entitled “Assessment of Glucose Lowering Medications effectiveness for cardiovascular clinical risk management of real-world patients with type 2 diabetes. Targeted maximum likelihood estimation under model misspecification and outcome missingness.”.

In the attached file you will find a list of changes related to the manuscript. We have provided point-by-point responses. The changes made in the manuscript are highlighted with track changes. Thus, all changes can be read in the manuscript as well. We used the attached document of manuscript, as recommended.

Reviewer 2 Report

The article addresses an important public health topic Type 2 Diabetes with a promising statistical methodology TMLE and combined with the missing data approach.

However, I consider that there is an indiscriminate use of statistical and computational techniques that makes it impossible to obtain clear conclusions about the applied procedures and their results. The following are some questions in this direction.

On page 3-What are the criteria to define directed acyclic graph (DAG), shown in Figure 1, which determines the dependencies between the variables? How is it justified, for example, not to assume the dependence of LDL with BMI?

In lines 159-160 one can read:” In the simulation study, we performed the comparisons in the most common real world situation: both the treatment and outcome models were” How was it determined that the models were misspecified? What measure or statistical criteria were used?.

In lines 230 Delta not set which makes reading difficult

On pages 6-7 many procedures and statistical models were applied however the justification of the choice was insufficiently justified.

In the context of the problem, what is the correctly specified model to evaluate the robustness of the proposal presented in relation to model misspecification? As this model probably cannot be defined, how do the presented techniques evaluate this? just for the bias

Why was the longitudinal character of the study not taken into account?

Author Response

(The authors gave the same response as above.)

Reviewer 3 Report

This is an interesting and generally well-written paper using data from an Italian multicenter retrospective study (DARWIN-T2D) to compare management of cardiovascular outcomes for type 2 diabetes patients treated with SGLT2i or GLP-1RA drugs. Missing data were imputed for all but the outcome variable and marginal treatment effects (MTEs) were estimated using logistic regression, multiple propensity score (PS) approaches, and targeted maximum likelihood estimation (TMLE). Simulation methods were used to compare bias and standard errors across methods under different assumptions regarding model misspecification and outcome missingness.

In contrast to the logistic regression and PS approaches, TMLE indicated that dapagliflozin was significantly more effective than GLP-1RA for management of CV risk among T2D patients. Moreover, the simulation results indicate that TMLE has the lowest bias and standard errors of MTE of any of the statistical estimators that were examined.

I have several relatively minor comments. The first related to terminology regarding treatment effects. Throughout the paper, the authors refer to the estimation of MTE. Three major variants of treatment effects are average treatment effects (ATE), average treatment effects of the treated (ATT), and marginal treatment effects (MTE).  Given that the estimates were based on patients treated with either SGLT2i or GLP-1RA drugs, it would seem that the estimates presented in the paper are actually ATTs. MTEs, on the other hand, are relatively new to the empirical literature and have important implications for choice of the statistical estimator. In particular, the estimation of MTEs highlights two key issues: (i) the existence of common support among the treatment groups; and (ii) the presence of unobserved essential heterogeneity in treatment selection and outcomes. Logistic regression takes account of neither common support nor unobserved treatment heterogeneity. PS methods and TMLE account for common support but not unobserved treatment heterogeneity. Basu et al., show that MTEs are the most general of the three treatment effects and that ATEs and ATTs can be derived from MTEs.  However, estimation of MTEs requires the use of local instrumental variable methods.

Basu A, Heckman J, Navarro-Lozano S, Urzua S. Use of instrumental variables in the presence of heterogeneity and self selection: an application to treatments of breast cancer patients.

Health Econ. 2007;16:1133–57.

My second comment refers to other literature utilizing simulation methods to compare the properties of TMLE to those of other statistical estimators. This literature has found similar evidence of the desirable statistical properties of TMLE and the current paper would benefit from briefly reviewing this literature. I am sure that there are others, but I am aware of the following three papers:

Kreif, N., Tran, L., Grieve, R., Stavola, B., Tasker, R., & Petersen, M. (2017).

Estimating the comparative effectiveness of feeding interventions in the pediatric

intensive care unit: A demonstration of longitudinal targeted maximum

likelihood estimation. American Journal of Epidemiology, 186, 1370–1379.

Pang, M., Schuster, T., Filion, K., Schnitzer, M., Eberg, M., & Platt, R. (2016).

Effect estimation in point-exposure studies with binary outcomes and high dimensional

covariate data - a comparison of targeted maximum likelihood

estimation and inverse probability of treatment weighting. The international

Journal of Biostatistics, 12.

Schuler, M., & Rose, S. (2017). Targeted maximum likelihood estimation for causal

inference in observational studies. American Journal of Epidemiology, 185,

65–73.

My third point has to do with the importance of research design for any discussion of bias in treatment effect estimates using observational data. I was happy to see the authors’ use of a DAG to generate their data for their simulations. To what extent did they use DAGs or the potential outcomes framework in their estimation of causal treatment effects? It is well known that errors in research design have been a major contributor to bias in previously published treatment effect estimates (see several works by Miguel Hernan and colleagues on this topic). Variation in bias by statistical estimator is a second order question, assuming an appropriate design. As such, any discussion of bias in treatment effect estimates should frame the discussion within a causal framework. This could be addressed by adding a sentence or two on the importance of design in observational studies and citing some of the paper by Hernan and colleagues.

Finally, although the writing in the manuscript is generally clear, there are a few instances of run-on sentences in the manuscript that would benefit from some editing.

Author Response

(The authors gave the same response as above.)
